# Cervical Cancer Stages, Human Papillomavirus Integration, and Malignant Genetic Mutations: Integrative Analysis of Datasets from Four Different Cohorts

**DOI:** 10.3390/cancers15235595

**Published:** 2023-11-26

**Authors:** Foziya Ahmed Mohammed, Kula Kekeba Tune, Marti Jett, Seid Muhie

**Affiliations:** 1Department of Software Engineering, College of Engineering, Addis Ababa Science and Technology University, Addis Ababa 16417, Ethiopia; foziya.ahmed@aastu.edu.et (F.A.M.); kula.kakeba@aastu.edu.et (K.K.T.); 2Center of Excellence for HPC and Big Data Analytics, Addis Ababa Science and Technology University, Addis Ababa 16417, Ethiopia; 3Department of Information Technology, College of Computing and Health Informatics, Wolkite University, Wolkite P.O. Box 07, Ethiopia; 4US Army Medical Research and Development Command, Head Quarter, Walter Reed Army Institute of Research, Silver Spring, MD 20910, USA; marti.jett-tilton.civ@health.mil; 5Medical Readiness Systems Biology, Walter Reed Army Institute of Research, Silver Spring, MD 20910, USA; 6The Geneva Foundation, Silver Spring, MD 20910, USA

**Keywords:** cervical cancer, human papillomavirus (HPV), productive integration, stages of cervical cancer, mutation, molecular pathways, protein networks

## Abstract

**Simple Summary:**

This study provides a thorough examination of various aspects of cervical cancer, using clinical and molecular datasets from four cohorts of patients across four continents. This integrative analysis explores the relationships between cervical cancer stages, productive human papillomavirus (HPV) integration, and the resulting malignant genetic mutations. The clinical data showed that cervical cancer and its stages were positively correlated with high-risk HPV infection and integration. At the molecular level, biological processes related to HPV infection, cancer-related conditions, and viral carcinogenesis were associated with the most significantly enriched pathways. The clinical and molecular findings further establish the prominent role of HPV infection and integration leading to genetic mutation underpinning cervical cancer and its stages across cohorts of different races and regions.

**Abstract:**

Cervical cancer represents a significant global health concern, stemming from persistent infections with high-risk types of human papillomavirus (HPV). The understanding of cervical cancer’s clinical correlates, risk factors, molecular mechanisms, stages, and associated genetic mutations is important for early detection and improved treatment strategies. Through integrated analysis of clinical and molecular datasets, this study aims to identify key factors that are overlapping and distinct across four cohorts of different races and regions. Here, datasets from four distinct cohorts of patients from Uganda (N = 212), the United States of America (USA) (N = 228), China (N = 106), and Venezuela (N = 858) were examined to comprehensively explore the relationships between cervical cancer stages, HPV types (clades), productive HPV integration, and malignant genetic mutations. Cohort-specific findings included the occurrence frequencies of cervical cancer stages and grades. The majority of patients from the USA and China were diagnosed with stages I and II, while those from Uganda were diagnosed with stages II and III, reflecting levels of awareness and the availability of HPV vaccines and screening services. Conversely, cervical cancer and its stages were positively correlated with HPV types (clades), HPV integration, and risk-factor habits across the cohorts. Our findings indicate that the more common squamous cervical cancer can be potentially due to productive HPV16 (clade 9) integration. At the molecular level, pathways related to HPV infection, cancer-related conditions, and viral carcinogenesis were among the most significant pathways associated with mutated genes in cervical cancer (across cohorts). These findings collectively corroborate the prominent role of HPV infection and integration leading to genetic mutation and hence to the development of cervical cancer and its stages across patients of distinct races and regions.

## 1. Introduction

Human papillomavirus (HPV) infection unequivocally stands as a primary causative factor in the development of cervical cancer. Persistent infection with high-risk HPV types, particularly HPV16 and HPV18, is strongly associated with the development of cervical cancer [1,2]. HPV persistence, even after cervical surgical excision of high-grade cervical lesions, is reported to be a recurrence risk factor [3]. For example, HPV persistence after primary conization is shown to increasingly correlate with the risk of developing recurrent high-grade cervical dysplasia for up to 1 year [4]. Earlier studies also showed the persistence of HPV as being an increased risk factor even beyond 12 months, particularly in older women with type 16 infection [5].

Integration of the HPV genome into host cell DNA is considered a driver mutation in cervical carcinogenesis, and it is a critical event in the progression from HPV infection to cervical cancer [6]. HPV integration disrupts the viral E2 gene, leading to uncontrolled expression of the viral oncogenes E6 and E7, which promote cellular transformation and malignant progression [7]. Different HPV types have varying oncogenic potential. High-risk HPV types, such as HPV16, HPV18, HPV31, HPV33, and HPV45, are associated with a higher risk of developing cervical cancer [1,8]. These HPV types belong to the 12 group 1A carcinogen HPV types (HPV16, HPV18, HPV31, HPV33, HPV35, HPV39, HPV45, HPV51, HPV52, HPV56, HPV58, and HPV59), which are grouped due to consistent and sufficient epidemiological, experimental, and mechanistic evidence of their carcinogenicity in humans for cervical cancer [9]. Ninety six percent of cervical cancers are attributable to one of the twelve most common HPV types (group 1A) and HPV68 (group 2A) [9]. The presence of specific HPV types can also influence the prognosis and response to treatment in cervical cancer [10]. HPV infection is typically detected in early-stage cervical lesions, such as low-grade squamous intraepithelial lesions and high-grade squamous intraepithelial lesions [11]. As the disease progresses, HPV integration becomes more prevalent, particularly in advanced stages of cervical cancer [7].

Cervical cancer caused by productive high-risk HPV integration is characterized by various genomic alterations [12,13]. Host-related genetic factors, including susceptibility loci, play a substantial role in cervical cancer [14]. Integration-dependent disruption of high-risk HPV E2 gene functions are important for neoplastic transformation in cervical carcinogenesis [15]. Also, the PI3K/AKT/mTOR pathway, mediated by ERBB3, plays a role in altering the epithelial–mesenchymal transition in cervical cancer [16]. Multi-region sequencing has depicted intratumor heterogeneity and clonal evolution in cervical cancer with frequently mutated genes. Somatically mutated human genes known to be sites for productive HPV integration in cervical cancer include MAPK1, HLA-B, EP300, FBXW7, NFE2L2, TP53, ERBB2, SHKBP1, ERBB3, CASP8, HLA-A, TGFBR2, FBXW7, PIK3CA, FAT1, TP63, MECOM, TBL1XR1, NDN, GOLGA6L4, BAIAP3, TTN, MUC4, KMT2C, and SYNE1 [12,17,18,19,20,21,22].

While HPV infection is a major risk factor for cervical cancer, a small proportion of cervical cancers are HPV-negative. HPV-negative cervical cancers may have distinct molecular characteristics and are associated with different risk factors, such as genetic alterations and other viral infections [23,24]. The presence of multiple high-risk factors can impact clinical outcomes and offset the benefits of certain treatments in early-stage cervical cancer [25,26]. Other risk factors, such as age at first intercourse, number of sexual partners, parity, race, socioeconomic status, and knowledge about cervical cancer screening, have also been associated with cervical cancer progression and severity [27,28]. It is important to note that the relationship among stages of cervical cancer, HPV infection, integration, and HPV types is a complex area of research, and further studies are needed to fully understand the underlying mechanisms and clinical implications.

Here, we examined data from four distinct cohorts of patients from Uganda, the USA, China, and Venezuela to identify overlapping and distinct relationships among cervical cancer stages, HPV types (clades), productive HPV integration, and malignant genetic mutations across patients of different races and regions. The 2020 estimated age-standardized incidence rates of cervical cancer in these countries per 100,000 women were 17.7 to <22.0 in Uganda, 8 to <11.2 in the USA, 11.2 to <14.8 in China, and 22.0 to <33.8 in Venezuela [29]. With regard to HPV vaccine coverages in these countries: (i) in the USA, according to the 2021 report on cancer-trend progress by the National Cancer Institute: 58.5% of adolescents aged 13–15 years had received two or three doses of HPV vaccines as recommended (https://progressreport.cancer.gov/prevention/hpv_immunization, accessed on 16 November 2023); (ii) in China, according to 2020 estimates, vaccination coverage was <1%, and most HPV vaccines were given to adult women rather than to young girls aged 9–14 years old, though there is a plan to reach 90% vaccination in girls by the age of 15 in the year 2030 [30]. In Northern Uganda, Lira City, it was reported that about 19.6% of schoolgirls aged 9–14 years had received the HPV vaccine [31], which is a tiny portion of the country. According to a 2022 evaluation, Venezuela is one of the countries that have not introduced the HPV vaccine as a public health policy [32].

The clinical and molecular datasets from the four cohorts provided valuable insights into the observational and correlative relationships among cervical cancer stages, HPV types (clades), genomic alterations, and the underlying biological pathways of cervical cancer across cohorts of patients belonging to distinct races and regions.

## 2. Materials and Methods

### 2.1. Datasets Used in This Study

Clinical, demographic, and molecular datasets from four cohorts of cervical cancer patients belonging to different races and regions were used in this study: (i) the TCGA dataset from the Cancer Genome Characterization Initiative, United States of America (USA); (ii) the MOCC dataset collected in China (from Chinese cervical cancer patients); (iii) datasets obtained from Ugandan cervical cancer patients; and (iv) datasets on cervical cancer risk factors collected at the ‘Hospital Universitario de Caracas’ in Caracas, Venezuela. Detailed descriptions of each of these datasets are given next.

Datasets generated by The Cancer Genome Atlas Research (TCGA) under the “Integrated genomic and molecular characterization of cervical cancer [17] (https://www.cancer.gov/tcga, accessed on 7 October 2023) and the Cancer Genome Characterization Initiative (https://ocg.cancer.gov/programs/cgci, accessed on 7 October 2023)” were accessed via the supplemental material of a *Nature Communications* paper [33]. The number of cervical cancer patients for the TCGA cohort, N = 228.Datasets from the MOCC (Chinese) cohort were obtained from the GitHub repository of the *Cell Genomics* paper “Multi-omics characterization of silent and productive HPV integration in cervical cancer” [34] (https://github.com/FlyPythons/MOCC/tree/v1.0.0/data, accessed on 7 October 2023). The number of cervical cancer patients, N = 106.Datasets for the Ugandan cohort were obtained from an online repository of the *Nature Genetics* paper “Analysis of Ugandan cervical carcinomas identifies HPV clade–specific epigenome and transcriptome landscapes” [35] (https://www.nature.com/articles/s41588-020-0673-7#Sec46, accessed on 7 October 2023). The number of cervical cancer patients, N = 212.Datasets on cervical cancer (risk factors) were collected at the ‘Hospital Universitario de Caracas’ in Caracas, Venezuela, and the datasets focus on the prediction of indicators or diagnosis of cervical cancer. These datasets comprise demographic information, habits, and historic medical records of 858 subjects [36] (https://archive.ics.uci.edu/dataset/383/cervical+cancer+risk+factors, accessed on 7 or 13 October 2023; DOI: 10.24432/C5Z310). The latter is licensed under a Creative Commons Attribution 4.0 International license (CC BY 4.0). Several patients decided not to answer some of the questions because of privacy concerns (missing values).

The cohorts from these countries were chosen mainly due to the availability of relevant and high-quality demographic, clinical, and molecular datasets that are representative of the major races and regions with divergent socioeconomic backgrounds.

Additionally, lists of somatically mutated genes associated with cervical cancer (collected from multiple data repositories or reported by other studies) were also searched and compared to the list of genes or proteins found at the Cervical Cancer Gene Database (CCDB) (https://webs.iiitd.edu.in/raghava/ccdb/stat.php, accessed on 13 October 2023).

### 2.2. Statistical and Machine Learning Methods, Gene Networks, and Visualization

Frequencies of cervical cancer stages across the datasets of different cohorts were calculated using epiDisplay [37] and visualized using ggplot2 [38], viridis [39], and hrbrthemes (https://github.com/hrbrmstr/hrbrthemes, accessed on 9 October 2023) libraries in the R programming language. Identification, ranking, and visualization of risk-factor predictors for cervical cancer were performed using the regularized random forest algorithm in R [40]. Correlations among cervical cancer risk factors, historical medical records, habits, and clinical features were identified using “Pearson correlation” in R with the option ‘pairwise.complete.obs’ and visualized using the corrplot [41] library in the R programming environment. Integrative analyses of clinical and molecular datasets within and across cohorts were carried out using custom R and python scripts along with multiple existing packages located at https://github.com/Foziyaam/CCclinicals. Mutated gene networks were constructed and significant pathways were enriched and visualized using the Reactome FI and BINGO packages of Cytoscape v3.9.1 and the Cytoscape visualization environment [42].

## 3. Results

Publicly available datasets from four different cohorts of cervical cancer patients (collected in Uganda, China, the USA, and Venezuela) were used for this study. The datasets from Uganda (N = 212), China (N = 106), and the USA (N = 228) include detailed clinical, demographic, and molecular features obtained from a total of 546 cervical cancer patients (Table 1). The features of the fourth dataset (from Venezuela, N = 858) are mainly cervical cancer risk factors, including historical medical records, habits, and demographic features (Appendix A). Integrative analysis of the first three datasets revealed conserved and distinct clinical and molecular features among cohorts of patients of different socioeconomic backgrounds, races, and regions.

### 3.1. Occurrence Frequencies of Cervical Cancer Stages across Three Different Cohorts

Clinical datasets from cervical cancer patients collected from three different regions (the USA, China, and Uganda) show distinct distributions of occurrence, reflecting the availability of screening and vaccination programs in each region (Figure 1). In the dataset from the USA, where there is better awareness, screening, medical services, and HPV vaccines, more than half of the patients were diagnosed with stage I cervical cancer, whereas the dataset from China shows that the majority of the patients were diagnosed with stages I and II (Figure 1). However, in the dataset from Uganda, the majority of patients were diagnosed with stages II and III, indicative of less awareness, screening, and medical services compared to the USA and China (Figure 1).

### 3.2. Clinical Features Conserved across Cohorts

Among the clinical features that were shared across cohorts, squamous histology was identified for the majority (70.9%) of the patients (Table 1), and it was the dominant histology across cohorts and stages of cervical cancer patients (Figure 2). Across cohorts, the majority of patients were HPV-positive (71.6%), the dominant type being HPV16 (43.3%), which belongs to clade A9 (49.3%) (Table 1), and HPV16 (clade A9) was dominant across the different stages of cervical cancer as well (Figure 2). Most of the HPV-positive patients had productive integration of the HPV virus (80.6%) (Table 1), and integration was proportionately distributed among patients of different stages (Figure 2). A minority of HPV-positive patients were also HIV-positive (28.3%) (Table 1), and many of the HIV-positive patients had stage II or III cervical cancer (Figure 2). The largest age groups of patients were the 41–50 and 51–60 age groups (Table 1). Blacks were the largest race affected by cervical cancer, comprising the largest numbers at stages II and III (Figure 2, Table 1). The black race tallied here includes the Ugandan cohort and the African-American group from the USA cohort (Table 1). The majority of Whites and Asians were diagnosed with stage I cervical cancer (Figure 2).

### 3.3. Correlation of Stages of Cervical Cancer with Clinical and Demographic Features

In the ranking of cervical cancer risk factors using the regularized random forest algorithm, HPV infection was found to be the topmost predictor (Figure 3a). This ranking was performed using the dataset collected at the ‘Hospital Universitario de Caracas’ in Caracas, Venezuela, consisting of demographic information, habits, and historic medical records of 858 patients. Consistent with the current literature, our machine learning prediction showed that HPV is the most important risk factor for cervical cancer development. This was further shown by the molecular evidence that productive integration of HPV leads to mutation of the corresponding genes, which makes it a key factor for cervical cancer development. This observation corroborates the World Health Organization’s Cervical Cancer Elimination Initiative (Appendix A) (credit: https://cceirepository.who.int/, accessed on 12 October 2023), wherein HPV vaccination is regarded as the primary prevention approach (Appendix A).

Correlation analysis among clinical features collected across three regions (the USA, China, and Uganda) from a total of 546 cervical cancer patients showed HPV integration to be directly correlated with HPV16 (clade A9) and HPV18 (clade A7) (Figure 3b). HPV integration was also directly correlated with the squamous type of cervical cancer, which in turn correlated with HPV16 (clade A9) (Figure 3b). These datasets show that more productive integration can potentially be due to the integration of HPV16 (clade A9) leading to the more common squamous cervical cancer. Integration of HPV18 (clade A7) has some positive correlation with adenosquamous cervical cancer (Figure 3b). Though not huge, there are some correlations of stage I with HPV16 (A9) and squamous cervical cancer; stage III with HPV45 (A7) and HIV status; and stage IV with HPV18 (A7) and neuroendocrine cancer (Figure 3b).

### 3.4. Biological Processes and Molecular Pathways Significantly Associated with Genes Mutated in Cervical Cancer

Among the significantly enriched biological processes and molecular pathways that were associated with mutated genes in cervical cancer were “human papillomavirus infection”, “cancer related conditions”, and “viral carcinogenesis” (Figure 4, Table 2). Many of these genes are also reported to be productive integration sites of HPV infection [18,20,21,22]. Other significant pathways include EGFR tyrosine kinase inhibitor resistance, cellular senescence, p53 pathway feedback loops 2, endometrial cancer, central carbon metabolism in cancer, microRNAs in cancer, the FoxO signaling pathway, and ErbB2/ErbB3 signaling events (Table 2). Significant enrichment for cancer-related conditions identified a number of mutated genes as being associated with an increased risk of malignant transformation (Figure 4b). Analysis of cancer drug targets identified four of the HPV-integration sites and mutated genes as targets of ten cancer drugs (Figure 4c).

## 4. Discussion

To our knowledge, this study leverages the most powered cervical cancer clinical and molecular datasets (N = 1404) collected from patients representing the major races of the world across four continents. Integrative analysis of the clinical and molecular datasets collected from the four different cohorts reveals both distinct and conserved clinical features and overlapping molecular pathways underpinning cervical cancer progression and severity. Distinct findings from the clinical datasets reflect the socioeconomic factors contributing to the occurrence frequencies of cervical cancer stages across different races and regions. Conserved clinical features were more or less consistent across cohorts of patients (regardless of race and region). Correlation of the squamous, the most common, cervical cancer type, with HPV16 (A9) integration was conserved across the cohorts, and this seems to indicate that the majority of HPV-caused cervical cancers are due to productive integration of HPV16. Integration of HPV16 and other HPV types has been shown to be responsible for the somatic mutation of genes shown to underlie the development of cervical tumorigenesis [17,33,34,35,43]. Productive HPV integration is associated with tumor aggressiveness and immunoevasion.

Our findings (across races and regions) are consistent with previous reports (from single-cohort studies) linking the persistent infection of high-risk HPV to the integration of HPV genes in the human genome [33,35]. Integration of high-risk HPV into the host genome is reported to cause instability of the host genome and plays a key driving role in the carcinogenesis of cervical cancer [43]. Such an unstable genomic state is a dominant feature of invasive cervical cancer. It has also been shown that high-risk-HPV is not only integrated with human genes at the DNA level, but also generates fusion products at the RNA level to produce virus–human gene fusion transcripts [43]. Highly expressed HPV–human fusion transcripts are the key drivers of cervical carcinogenesis and downregulate the transcription of tumor suppressor genes [43]. Fusion transcripts are prevalent in cervical cancer cases infected with HPV16/18 [43]. The functional consequences of HPV fusion transcripts for the biology and pathophysiology of HPV-driven cervical cancers suggest that productive HPV integration should be evaluated as an indicator of high risk for progression to aggressive cancers [34].

Pathways significantly associated with mutated genes were also found to corroborate the roles of the corresponding proteins in productive HPV integration, viral carcinogenesis, and cancer-related conditions. Multi-omics data obtained by HPV capture sequencing, RNA sequencing, and whole-genome bisulfite sequencing showed that patients at clinical stage II had the highest number of HPV integrations [44]. The phenomenon of co-localization of HPV integration events and mutations may lead to changes in crucial cancer gene functions [43]. HPV clade-specific differences were also detected in tumor DNA methylation, promoter- and enhancer-associated histone marks, gene expression, and pathway dysregulation [35]. Changes in histone modification in HPV integration events were correlated with upregulation of nearby genes and endogenous retroviruses [35]. Analysis of Ugandan cervical carcinomas identified HPV clade-specific epigenome and transcriptome landscapes [35].

This study and numerous other studies show that productive HPV integration into the host genome is a dominant feature of invasive cervical cancer [45]. Finally, it has to be made clear that most but not all cervical cancer patients were HPV-positive. For example, a unique set of endometrial-like cervical cancers, comprised predominantly of HPV-negative tumors, were reported with relatively high frequencies of KRAS, ARID1A, and PTEN mutations [17].

## 5. Conclusions

In conclusion, our integrative analyses of clinical and molecular datasets across diverse cohorts uncovered distinct and conserved features in cervical cancer. The frequencies of cancer stages reflected the accessibility of early screening services and HPV vaccines. Notably, features such as squamous cervical cancer type and productive HPV16/18 integration consistently dominated across cohorts. Somatic genetic mutations, associated protein networks, and pathways related to HPV infection and cancer were conserved across patients, underscoring their significance. Overall, cervical cancer and its stages demonstrated positive correlations with HPV types (clades), integration, and are crucial risk factors. For example, the prevalence of squamous cervical cancer potentially stemmed from productive HPV16 (clade 9) integration. Molecular analyses underscored the enrichment of pathways associated with HPV infection, cancer-related conditions, and viral carcinogenesis, collectively emphasizing the pivotal role of HPV infection and integration in genetic-mutation-mediated cervical cancer development across diverse populations.

## Figures and Tables

**Figure 1 cancers-15-05595-f001:**
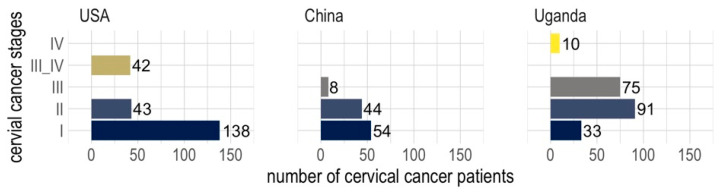
Occurrence frequencies of cervical cancer stages across three cohorts of patients from the USA, China, and Uganda.

**Figure 2 cancers-15-05595-f002:**
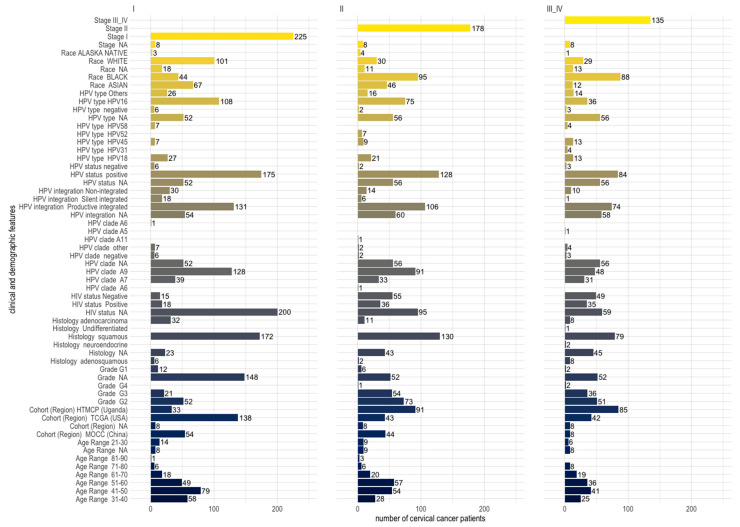
Distribution of clinical and demographic features across stages I, II, and III and IV of 546 cervical cancer patients (comprising three different cohorts from the USA, China, and Uganda). Patients belonging to the 31–40 and 41–50 age groups are the majority at stage I, while patients of the 51–60 age group are the majority at stage II.

**Figure 3 cancers-15-05595-f003:**
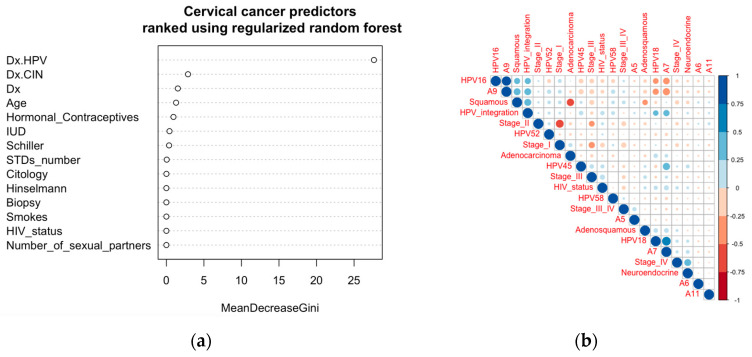
Ranking of risk factors as predictors and clinical and demographic correlates. (**a**) Predictors (risk factors) for cervical cancer identified using the regularized random forest algorithm and ranked according to the MeanDecreaseGini. (**b**) Correlations among clinical features and HPV types with the stages of cervical cancer. Similar patterns were observed in the dataset from the Venezuelan cohort (Appendix A).

**Figure 4 cancers-15-05595-f004:**
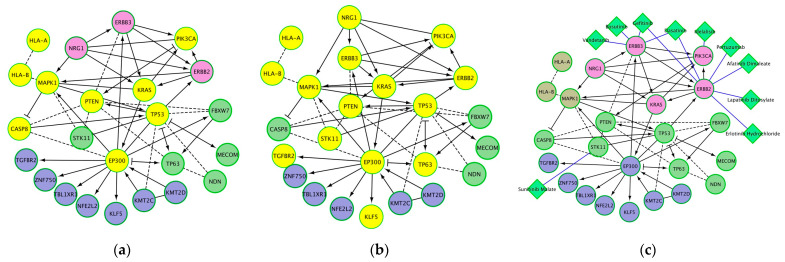
Protein networks and significantly associated pathways for genes that are sites of HPV integration and mutated in cervical cancer: (**a**) proteins significantly associated with human papillomavirus infection (yellow nodes); (**b**) proteins significantly associated with cancer-related conditions (yellow nodes); (**c**) anti-cancer drugs (diamond-shaped green nodes). Other nodes (besides the yellow colored nodes in (**a**,**b**) are shown to have functional and regulatory relations with the yellow colored nodes. Note: A cancer-related condition is a disorder either associated with an increased risk of malignant transformation (e.g., intraepithelial neoplasia, leukoplakia, dysplastic nevus, and myelodysplastic syndrome) or that develops as a result of the presence of an existing malignant neoplasm (e.g., paraneoplastic syndrome).

**Table 1 cancers-15-05595-t001:** Distribution of clinical and demographic features across three cohorts. Features were from 546 cervical cancer patients combined from the USA, China, and Uganda cohorts. Note: NA means either the patient did not give the information or it was not measured (missing).

Cervical Cancer Stage	I	II	III	III/IV	IV	NA			
	225	178	83	42	10	8			
Histology	Adenocarcinoma	Adenosquamous	Neuroendocrine	Squamous	Undifferentiated	NA			
	51	16	2	387	1	89			
Cancer Grade	G1	G2	G3	G4	NA				
	21	176	113	3	233				
HPV Status	Negative	Positive	NA						
	11	391	144						
HPV Type	HPV16	HPV18	HPV26	HPV30	HPV31	HPV33	HPV34	HPV35	
	220	62	1	2	11	9	1	2	
	HPV39	HPV45	HPV51	HPV52	HPV56	HPV58	HPV59	HPV6	
	3	29	1	13	2	14	6	1	
	HPV66	HPV68	HPV69	HPV70	HPV73	HPV82	HPV9	Negative	NA
	2	3	2	1	2	3	1	11	144
HPV Clade	A5	A6	A7	A9	A11	Negative	Other	NA	
	1	2	104	269	1	11	14	144	
HPV Integration	Non-integrated	Productive integrated	Silent integrated	NA					
	54	315	25	152					
HIV Status	Negative	Positive	NA						
	122	89	335						
Age Range	21–30	31–40	41–50	51–60	61–70	71–80	81–90	NA	
	29	111	174	145	59	22	5	1	
Age Stat	Min.	1st Qu.	Median	Mean	3rd Qu.	Max.	NA		
	21	40.24	48	49.01	56.71	89	1		
Race	Alaska Native	Asian	Black	White	NA				
	8	125	231	163	19				
Cohort (Region)	TCGA (USA)	MOCC (China)	HTMCP (Uganda)						
	228	106	212						

**Table 2 cancers-15-05595-t002:** Significant pathways associated with genes that are sites of HPV integration and mutated in cervical cancer.

Significantly Enriched Pathway	Proteins from Network	False Discovery Rate	Nodes
EGFR tyrosine kinase inhibitor resistance	7	1.01 × 10^−7^	PTEN, ERBB3, ERBB2, MAPK1, NRG1, PIK3CA, KRAS
Human T-cell leukemia virus 1 infection	9	1.01 × 10^−7^	PTEN, MAPK1, EP300, HLA-B, HLA-A, TGFBR2, PIK3CA, KRAS, TP53
Cellular senescence	8	1.31 × 10^−7^	PTEN, MAPK1, HLA-B, HLA-A, TGFBR2, PIK3CA, KRAS, TP53
Prostate cancer	7	1.31 × 10^−7^	PTEN, ERBB2, MAPK1, EP300, PIK3CA, KRAS, TP53
p53 pathway feedback loops 2	5	1.74 × 10^−7^	PTEN, TP63, PIK3CA, KRAS, TP53
Endometrial cancer	6	1.88 × 10^−7^	PTEN, ERBB2, MAPK1, PIK3CA, KRAS, TP53
Kaposi sarcoma-associated herpesvirus infection	8	2.26 × 10^−7^	CASP8, MAPK1, EP300, HLA-B, HLA-A, PIK3CA, KRAS, TP53
Pathways in cancer	11	2.35 × 10^−7^	PTEN, CASP8, MECOM, ERBB2, MAPK1, EP300, TGFBR2, PIK3CA, KRAS, TP53, NFE2L2
Viral carcinogenesis	8	3.15 × 10^−7^	CASP8, MAPK1, EP300, HLA-B, HLA-A, PIK3CA, KRAS, TP53
Central carbon metabolism in cancer	6	3.15 × 10^−7^	PTEN, ERBB2, MAPK1, PIK3CA, KRAS, TP53
MicroRNAs in cancer	9	3.43 × 10^−7^	PTEN, ERBB3, ERBB2, MAPK1, EP300, TP63, PIK3CA, KRAS, TP53
FoxO signaling pathway	7	3.43 × 10^−7^	PTEN, STK11, MAPK1, EP300, TGFBR2, PIK3CA, KRAS
Chronic myeloid leukemia	6	3.94 × 10^−7^	MECOM, MAPK1, TGFBR2, PIK3CA, KRAS, TP53
Human papillomavirus infection	9	4.65 × 10^−7^	PTEN, CASP8, MAPK1, EP300, HLA-B, HLA-A, PIK3CA, KRAS, TP53
ErbB2/ErbB3 signaling events	5	4.65 × 10^−7^	ERBB3, ERBB2, MAPK1, NRG1, KRAS

## Data Availability

Custom R and python scripts and detailed analysis steps are available at https://github.com/Foziyaam/CCclinicals.

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
