# Peer review of "Cervical Cancer Stages, Human Papillomavirus Integration, and Malignant Genetic Mutations: Integrative Analysis of Datasets from Four Different Cohorts"

_cancers, 2023, doi:10.3390/cancers15235595_

Round 1
Reviewer 1 Report
Comments and Suggestions for Authors
I read with great interest the Manuscript titled " Is “Cervical cancer stages, human papillomavirus integration and malignant genetic mutations: integrative analysis of datasets from four different cohorts" which falls within the aim of the Journal.
Although the manuscript can be considered already of good quality, I would suggest following recommendations:
- I suggest round of language revision, in order to correct few typos and improve readability;
- Considering topic and results if this study, I suggest that authors to add a reference to current evidence about crucial role of HPV infection persistence, also after treatment of cervical lesions, and its consequences. (for example I would be glad if the authors discuss this important point PMID: 37401466 and 37747763.
Because of these reasons, the article should be revised and completed. Considering all these points, I think it could be of interest to the readers and, in my opinion, it deserves the priority to be published after minor revisions
Comments on the Quality of English Language
I suggest round of language revision, in order to correct few typos and improve readability
Reviewer 2 Report
Comments and Suggestions for Authors
This study aimed to 'through integrated analysis of clinical and molecular da- 28 tasets, this study aims to identify key factors that are overlapping and distinct across the four cohorts of different races and regions'. It is an interesting study.
In the introduction section the authors should add incidence rates of cervical cancer in US, China, Uganda and Venezuela.
The authors should explain why the cohorts from these particular countries were chosen?
Was the information regarding HPV vaccination available from these cohorts?
In the discussion section the authors should further justify why this analysis was conducted and what are the possible clinical implications (if any) of this study.
Reviewer 3 Report
Comments and Suggestions for Authors
Mohammed et al. have conducted an extensive analysis of cervical cancer, a pressing global health issue primarily linked to persistent human papillomavirus (HPV) infections. The study delves into clinical and molecular aspects, including stages, risk factors, and genetic mutations, across four diverse cohorts from Uganda, the United States of America (USA), China, and Venezuela.
In this integrative analysis of datasets encompassing 212 patients from Uganda, 228 from the USA, 106 from China, and 858 from Venezuela, the authors aim to identify commonalities and distinctions in factors contributing to cervical cancer across different racial and regional contexts. The study reveals variations in the distribution of cervical cancer stages and grades among cohorts, with higher incidences of stages I & II in the USA and China, potentially reflecting differences in awareness, HPV vaccination availability, and screening services. Conversely, Ugandan patients exhibited a higher prevalence of stages II & III.
The investigation highlights positive correlations between cervical cancer and its stages with HPV types (clades), productive HPV integration, and specific risk factors across all cohorts. Notably, the study emphasizes a potential association between the more prevalent squamous cervical cancer and productive HPV16 (clade 9) integration.
At the molecular level, the research identifies significant pathways related to HPV infection, cancer-related conditions, and viral carcinogenesis associated with mutated genes in cervical cancer across diverse cohorts. These findings collectively underscore the pivotal role of HPV infection and integration in driving genetic mutations that contribute to cervical cancer development across patients from distinct races and regions. The integrated approach provides valuable insights for early detection and the development of improved treatment strategies for cervical cancer. Overall, the study concludes that cervical cancer and its stages are consistently positively correlated with HPV types (clades), integration, and crucial risk factors, consolidating the prominent role of HPV infection and productive integration leading to genetic mutation in mediating the development of cervical cancer across diverse populations.
The claims are properly placed in the context of the previous literature. The experimental data support the claims. The manuscript is written clearly enough that most of it is understandable to non-specialists. The authors have provided adequate proof for their claims, without overselling them. The authors have treated the previous literature fairly. The paper offers enough details of methodology so that the experiments could be reproduced.
Minor revisions
Line 15, Simple Summary, "This study provides a thorough examination of various aspects of cervical cancer, utilizing clinical and molecular datasets from four cohorts of patients across four continents."
Line 25, Abstract, "Cervical cancer represents a significant global health concern, stemming from persistent infections with high-risk types of human papillomavirus (HPV)."
Line 49, Introduction, "Human papillomavirus (HPV) infection unequivocally stands as a primary causative factor in the development of cervical cancer."
Line 56-57, "High-risk HPV types, such as HPV16, HPV18, HPV31, HPV33, and HPV45 are associated with a higher risk of developing cervical cancer."
Tjalma WA, Depuydt CE. Don't forget HPV-45 in cervical cancer screening. Am J Clin Pathol. 2012 Jan;137(1):161-2; author reply 162-3. doi: 10.1309/AJCPYB6C4HIMLZIX. PMID: 22180491.
https://pubmed.ncbi.nlm.nih.gov/22180491/
Arbyn M, Tommasino M, Depuydt C, Dillner J. Are 20 human papillomavirus types causing cervical cancer? J Pathol. 2014 Dec;234(4):431-5. doi: 10.1002/path.4424. PMID: 25124771.
https://pubmed.ncbi.nlm.nih.gov/25124771/
Line 185, Table 1, "I would like to bring to your attention some formatting concerns in Table 1. Specifically, entries for HPV 33, HPV 34, HPV 58, HPV 59, and HPV 82 appear to be divided across two separate lines, which makes it challenging to read and identify the complete HPV numbers. Please consider adjusting the formatting for a clearer presentation of the information related to these HPV types."
Line 289-307, Conclusions, "In conclusion, our integrative analyses of clinical and molecular datasets across diverse cohorts uncovered distinct and conserved features in cervical cancer. The frequency of cancer stages reflected the accessibility of early screening services and HPV vaccines. Notably, features such as squamous cervical cancer type and productive HPV16/18 integration consistently dominated across cohorts. Somatic genetic mutations, associated protein networks, and pathways related to HPV infection and cancer were conserved across patients, underscoring their significance. Overall, cervical cancer and its stages demonstrated positive correlations with HPV types (clades), integration, and crucial risk factors. For example, the prevalence of squamous cervical cancer potentially stemmed from productive HPV16 (clade 9) integration. Molecular analyses underscored the enrichment of pathways associated with HPV infection, cancer-related conditions, and viral carcinogenesis, collectively emphasizing the pivotal role of HPV infection and integration in genetic mutation-mediated cervical cancer development across diverse populations."
Line 318, Figure S2, "Citology" => "Cytology"
